# Supplement Consumption and Periodontal Health: An Exploratory Survey Using the BigMouth Repository

**DOI:** 10.3390/medicina59050919

**Published:** 2023-05-11

**Authors:** Muhammad H. A. Saleh, Ann Decker, Mustafa Tattan, Omar Tattan, Joseph Decker, Abdusalam Alrmali, Hom-Lay Wang

**Affiliations:** 1Department of Periodontics and Oral Medicine, University of Michigan School of Dentistry, Ann Arbor, MI 48109, USA; 2Department of Periodontics, University of Iowa College of Dentistry, Iowa City, IA 52242, USA; 3RAK College of Dental Sciences, RAK Medical & Health Sciences University, Ras Al-Khaimah 11172, United Arab Emirates; 4Department of Cariology, Restorative Sciences, and Endodontics, University of Michigan School of Dentistry, Ann Arbor, MI 48109, USA

**Keywords:** periodontal, periodontitis, diseases, risk factor assessment, demographic survey

## Abstract

Background: Dietary supplements have been investigated for their impact on the periodontal apparatus (alveolar bone, mucosa, periodontal ligament, and cementum) and their hypothetical protective role against periodontitis. There remains a gap in the field in this area. Thus, the present study aims to examine the correlation between populations who report taking different dietary supplements and their relative periodontal health. Methods: The BigMouth dental data repository derived from the dental Electronic Health Records (EHRs) of the University of Michigan school of dentistry was used to extract data relating to all patients who fulfilled the eligibility criteria. The prevalence of periodontitis compared to periodontal health as related to supplement consumption was assessed. Results: A total of 118,426 individuals (55,459 males and 62,967 females) with self-reported consumption of the dietary supplements of interest were identified in the University of Michigan database via the BigMouth repository. Associations with the following vitamins were investigated, Vitamin B, Vitamin C, Vitamin D, Vitamin E, Multivitamins, Fish oil, Calcium, Omega 3, Saw palmetto, Zinc, Sildenafil, Flax seed, Folic acid, Garlic pills, Ginger pills, Ginko, Ginseng, Glucosamine, Iron, and Magnesium. Out of these supplements, only multivitamins and iron were found to significantly favor periodontal health, while folic acid and vitamin E significantly favored periodontitis. Conclusions: This study found a minimal association between the consumption of dietary supplements with periodontal health.

## 1. Introduction

In today’s health-conscious world, many individuals embracing a fitness lifestyle turn to dietary supplements and so-called superfoods to enhance their well-being, overall fitness, and recovery [1]. These products are believed to impact various tissues and processes that could also be relevant to periodontal healing and regeneration [2,3]. While there is a growing body of research exploring the impact of these supplements and superfoods on human health and their potential medical applications, limited information is available regarding their effects on periodontal tissues and their potential use in periodontal treatment or medicine [2,4,5]. For instance, a recent meta-analysis examined the role of spirulina, a well-known superfood, in reducing oxidative stress and inflammation, finding significant improvements in biomarkers [6]. Similarly, a meta-analysis by demonstrated the beneficial effects of probiotics on gut health and their potential to improve mental health outcomes6. Furthermore, the efficacy of curcumin, a bioactive compound found in turmeric, in managing chronic inflammatory conditions, such as rheumatoid arthritis was investigated due to its potent anti-inflammatory properties and found to provide a profound analgesic and anti-inflammatory effect [7].

The host response involves innate immune cells including polymorphonuclear leukocytes (PMNs), which serve as a first line of defense against invading periodontal pathogens [8]. After stimulation by microbiological proteins, PMNs produce robust quantities of reactive oxygen species (ROS), hypochlorous acid (HOCl), and other inflammatory molecules during bacterial phagocytosis [9]. These biological products can cause unintentional damage to the surrounding tissue by lipid peroxidation, oxidation of other enzymes (i.e., antiproteinases), and damaging DNA or extracellular matrix [9]. In addition to host damage, bacterial products propagate a positive feedback loop of proinflammatory cellular migration and cytokine release by monocytes and macrophages [9].

Many authors have hypothesized that rebalancing metabolic cofactors and environmental nutrients may aid in mitigating the inflammatory cascade in periodontal disease [10,11]. Macro- and micro- nutrient deficiencies may interfere with the human body’s ability to cope with infection or byproducts of bacterial phagocytosis and propagate exacerbated host responses contributing to periodontal disease onset and progression mechanisms [10]. Dietary supplements have been previously investigated for their impact the periodontal apparatus, (alveolar bone, mucosa, periodontal ligament, and cementum) [1,11].

Supplementation studies provide an important assessment of the systemic landscape that influences inflammatory diseases, including periodontitis; however, there remains a gap in the field in this area. For instance, suggest that Vitamin D/Calcium supplementation may alveolar resorption and tooth loss; an large epidemiological study revealed significant associations between periodontal health and vitamin D and calcium intakes [12]. In addition, a one year prospective cohort study showed a modest positive effect on periodontal health, but noted dually that consistent dental care improved parameters of periodontal disease in all groups regardless of supplement intake [13].

Similarly, a recent investigation by Zong and colleagues discovered a negative correlation between blood vitamin B12 levels and the severity of periodontitis [14]. Earlier research has also suggested a potential connection between low levels of vitamin B12 in the serum and periodontitis [15]. Moreover, decreased blood vitamin B9 concentrations have been noticed in smokers, which might contribute to periodontitis [16,17]. Nonetheless, the underlying cause of this relationship remains uncertain. In a clinical trial involving 30 participants, Neiva et al. found that supplementation with B-vitamin complex promoted healing following periodontal flap surgery [18].

Thus, the purpose of the present study is to examine the correlation between populations who report taking different dietary supplements and their relative periodontal health.

## 2. Methods

This study was conducted in agreement with the Helsinki Declaration of 1975 (World Medical Association, 1975) as most recently revised in 2013 (World Medical Association, 2013). This retrospective case-control study was also in accordance with the STROBE (Strengthening the reporting of observational studies in epidemiology) checklist. The study was also approved by the University of Michigan Medical School Institutional Review Board (IRBMED) (identifier number: HUM00157260; Amendment: Ame00122603). Data were collected before the latest data load, May 2022.

### 2.1. Predetermined Definitions

Periodontitis: a category of patients determined based on the presence of planned or completed current CDT codes that resemble surgical and non-surgical treatment of periodontitis in the respective patient’s EHR (e.g., D4341, D4342, D4240, D4241, D4260, D4261, etc.)Periodontal health/Gingivitis: a category of patients determined based on the presence of planned or completed dental terminology (CDT) codes that resemble treatment–or lack thereof–consistent with periodontal health and/or gingivitis in the respective patient’s EHR (e.g., D1110) in addition to absence of any codes assigned to the periodontitis group.

### 2.2. Sample Selection

The present data was extracted from the centralized dental data repository known as BigMouth [19]. The data extraction, analysis and reporting of this investigation are based on level 1 data access. This included data pertaining to all patients who fulfilled the eligibility the criteria and were seen between 2011–2022 (±6 months, since the data repository is updated biannually) specifically at the University of Michigan School of Dentistry (Ann Arbor, MI, USA).

The general eligibility criteria for the sample were dentate adult patients (≥18 years old) receiving either preventative or periodontal treatment.

### 2.3. Data Extraction

A broadly inclusive set of queries were generated to identify count data related to periodontal health and self-reported consumption of dietary supplements based on the questionnaire patient are required to fill at the initial visit and updated every 6–12 months. Based on the available repository data, the investigated dietary supplements were vitamins B, C, D and E, multivitamins, fish oil, calcium, omega 3, saw palmetto, sildenafil and tadalafil, zinc, flax seed, folic acid, garlic pills, ginger pills, ginko, ginseng, glucosamine, iron and magnesium.

The queries, based on CDT codes in patient EHRs, are outline below:#1—“D4210” OR “D4211” OR “D4240” OR “D4241” OR “D4245” OR “D4260” OR “D4261” OR “D4263” OR “D4266” OR “D4274” OR “D4341” OR “D4342” OR “D4910”#2—“D1110”#3—#2 NOT #1

Thus, the periodontitis and periodontal health/gingivitis groups were all the patients who fit into the results of query #1 and #3, respectively. The presence of self-reported supplements was detected algorithmically.

### 2.4. Data Analysis

Summary statistics were calculated as counts and percentages for all categorical variables. These comprised frequency data representing prevalence of periodontal, systemic, and dietary supplements (e.g., periodontitis, folic acid, multivitamins, etc.) (Table 1). Chi-square test of independence was used to identify the presence of a relationship between periodontal diagnosis and presence of self-reported dietary supplement(s). Statistical significance was set at a *p*-value of 0.05. All analyses were performed using a dedicated software.

## 3. Results

### 3.1. Sample Characteristics

The overall study sample comprised 118,524 individuals (55,459 males and 62,967 females) (Table 1). A total of 85,809 (72.4%) individuals fit into the periodontal health/gingivitis definition and 32,715 (27.6%) into the periodontitis definition. The highest prevalence of any dietary supplement was that of multivitamins (5945 with periodontal health and 2073 with periodontitis) and vitamin D (4726 with periodontal health and 2025 with periodontitis), followed by calcium (2780 with periodontal health and 1198 with periodontitis) and fish oil (2200 with periodontal health and 939 with periodontitis).

### 3.2. Periodontal Conditions and Supplement Intake

The data analysis results are summarized in Table 1 and Table 2. The correlation of a total of 21 supplements to periodontitis/periodontal health was assessed. Four supplements, namely multivitamins, iron, folic acid and vitamin E, demonstrated a statistically significant greater frequency in one periodontal status category versus the other (*p* < 0.05). While consumption of multivitamins and iron significantly favored periodontal health, consumption of folic acid and vitamin E significantly favored periodontitis. The supplements magnesium and sildenafil were barely short of statistically significantly in favor of periodontal health (*p* = 0.055) and periodontitis (*p* = 0.068), respectively.

### 3.3. Gender Differences

The data analysis could not indicate gender differences for all supplements due to missing data (Table 1). The results were as follows for patients who had gender data available 12.3% for Vitamin B (2961 participants, 592 males and 1471 females), 10.1% for Vitamin C (2138 participants, 550 males and 983 females), 29.4% for Vitamin D (6751 participants, 1261 males and 3463 females), 3.9% for Vitamin E (881 participants, 191 males and 399 females), 25.7% for multivitamins (8018 participants, 2236 males and 3707 females), 21.3% for fish oil (3139 participants, 894 males and 1306 females), 16.4% for calcium (3978 participants, 413 males and 2367 females), and 4.2% for Omega 3 (797 participants, 193 males and 373 females).

## 4. Discussion

Our current study examined the correlation between a diagnosis of periodontitis and chart reporting of dietary supplementation. These correlations were able to capture associations between periodontitis and several vitamin and mineral supplements, including vitamin E, iron, folic acid, or a nonspecific multivitamin, however the nature of the level one BigMouth data limits our ability to associate these findings with different patient populations. Many of these correlations have been previously explored by other authors that are specific to different populations. For example, one report in the literature established notable differences in populations that take supplementation and seek care at a periodontal clinic [11]. These authors reported greater supplement use by females compared to males [11]. In addition younger patients (ages 31–50 years) had the highest frequency of no supplement use compared to patients over the age of 50 had the highest frequency of using greater or equal to four supplements [11]. Also of note, the study reported that smokers had lower frequency of supplement use compared to non-smokers [11]. Importantly, our level one study data does not allow stratification by patient, but rather only offers the removed, global view without control for stratification by these noted population in the literature. As such, in the future at the level two analysis, we intend to correct for these controls; however, at the level one stage it is outside the scope of the present study.

The present study data analysis reported patients taking a multivitamin were less likely to also have a diagnosis of periodontitis. Several articles reported evidence that multivitamin supplementation, specifically the antioxidant components within the multivitamin, have a modest, albeit significant effect on reduction of gingival inflammation [10,20,21,22]. One study reported that use of a multivitamin along with as several phytonutrients demonstrated reduction in gingival inflammation within the period of evaluation (8 weeks), but no significant effects on probing depth or clinical attachment levels compared to controls were observed [22]. Though it is possible the additive components of the multivitamin might ensure adequate nutrition important for overall wellness and periodontal health, contribution of the different classes might be needed in different populations, and this is an additional line of future investigation that is needed.

Vitamin E was identified in our study as significantly associated with a diagnosis of periodontitis. Vitamin E is a plant-based lipophilic vitamin that exhibit antioxidant and anti-inflammatory properties. Previous reports have hypothesized that vitamin E could improve the periodontal status by correcting redox status imbalance, reduce inflammatory response, and promote wound healing [23]. However, direct evidence for use of vitamin E supplementation on periodontal disease is still limited. One article reported that vitamin E may have the potential to reduce oxidative damage, but does not prevent alveolar bone loss in a controlled animal model [24]. The present study found that patients taking vitamin E supplements were more likely to have a concomitant diagnosis of periodontal disease. However, there was no ability for us at this stage and scope of the study to stratify patients taking vitamin E by sub-population or assess serum levels. Towards this point it is highly possible that the population taking vitamin E could be an aged population with additional inherent risk for periodontal disease due to its popular use in anti-aging applications [25,26]. As such, patient level data is required for patient stratification and controls of confounding variables to clearly assess the association reported.

Patients taking folic acid also experienced an increased risk of also having diagnosis of periodontal disease in our dataset. Folic acid is a member of the of the B-complex family and facilitates the mechanisms of cell division and cellular metabolism, as well as some immune cell functions. Folic acid deficiency surfaces with a similar clinical presentation to vitamin B12 deficiency. This in mind, supplementation with folic acid specifically is typically prescribed for deficiency or increased demand for the vitamin (i.e., pregnancy) [27]. Notably early pregnancy levels of gingival crevicular fluid matrix metalloproteinases-8 and -9 were associated with severity of periodontitis and development of gestational diabetes [28]. Because the current analysis lacks the ability to discern patient-level statistics and control conditions, as such to understand these associations further level-two, patient-level statistics can be performed in the future.

In the present study, patients taking iron supplements were decreased risk of periodontal disease. Recently, a meta-analysis reported that periodontitis Hb concentration disturbs the balance of iron metabolism and association with anemia inflammation [29]. Further, iron-deficiency anemia can lead to reduction of antioxidant enzymes, leading to an increased oxidative stress and worsening of periodontal diseases, but no studies have attempted to find a correlation between iron supplementation and outcomes of periodontal therapy [30]. The present study is starting the assessment of the impact that iron supplementation has on periodontal disease progression and impact on therapeutic outcomes; however, the limitation of this analysis includes assessment of serum levels prior to supplementation or at the time of periodontal assessment and controls of other related co-factors. Thus, we aim to investigate the relationship of iron supplementation and periodontal disease progression with increased rigor at the patient level.

Our analysis of the BigMouth dataset revealed interesting correlations between some known factors contributing to periodontal disease yet did not find significance in others. For example, vitamin D and calcium are essential components of the bone mineralization, and have been previously examined for their effects on periodontal disease [12,31]. Chronically low vitamin D levels leads to a low calcium balance, which in turn catalyzes bone loss and an osteoporosis phenotype. In addition to Vitamin D’s role in bone mineralization, the active form (1a, 25-dihydroxyvitamin) has been identified as an anti-inflammatory immunomodulatory molecule, capable of suppressing cytokine expression of monocytes and macrophages [32]. Some studies suggest that Vitamin D/Calcium supplementation reduced tooth loss and alveolar resorption and a large NHANES III study of twelve thousand adults revealed significant associations between periodontal health and vitamin D and calcium intakes [12,33]. In addition, a one year prospective cohort study showed a modest positive effect on periodontal health, but noted dually that consistent dental care improved parameters of periodontal disease in all groups regardless of supplement intake [13]. In addition, the study reported inclusion of only males aged 50–80 years and post-menopausal females [13]. However, one group conducted a clinical study and reported that calcium supplementation did not affect periodontal parameters [34]. The lack of a definitive consensus on the effects of Vitamin D and Calcium supplementation could be attributed to the heterogeneity in study populations and lack of control for serum concentration assessment, age, or other confounding biological factors. In our level one retrospective review, Vitamin D and Calcium intake did not have a significant impact on prevalence of periodontal disease, contributing to this body of literature.

As with all supplementations, understanding of the entire systemic landscape is crucial to assessment of disease onset and disease. The present study is limited by the information available by the retrospective data collection including lack of serum data and controls for known cofactors of disease onset/progression yet provides important insights into the broader populations associated with dietary supplementation and periodontitis. Addressing these limitations are currently outside the scope of this study but are important considerations we aim to address in the next, level-two studies.

## 5. Conclusions

Employing the BigMouth dental data repository, this study suggests a limited association between periodontal health and supplement intake. Associations with Vitamin B, Vitamin C, Vitamin D, Vitamin E, Multivitamins, Fish oil, Calcium, Omega 3, Saw palmetto, Zinc, Sildenafil, Flax seed, Folic acid, Garlic pills, Ginger pills, Ginko, Ginseng, Glucosamine, Iron, and Magnesium. Out of these supplements, only multivitamins and iron were found to significantly favor periodontal health, while folic acid and vitamin E significantly favored periodontitis.

## Figures and Tables

**Table 1 medicina-59-00919-t001:** Supplement consumption prevalence with gender predilection in periodontal health as compared to periodontitis.

	Conditions	Total (*n*)	Total Periodontal Health/Gingivitis (*n*)	Percent (+) for Supplement Consumption with Periodontal Health/Gingivitis	Gender (M/F)	Total Periodontitis (*n*)	Percent (+) for Supplement Consumption with Periodontitis	Gender (M/F)	*p*-Value *
Periodontal status only			85,809	-	38,417/47,313	32,715	-	17,042/15,654	-
Dietary Supplements	Vitamin B	2961	16,796	12.3%	592/1471	7027	12.8%	329/568	0.320
	Vitamin C	2138	15,217	10.1%	550/983	6341	9.5%	236/369	0.243
	Vitamin D	6751	16,056	29.4%	1261/3463	6682	30.3%	649/1375	0.196
	Vitamin E	881	14,991	3.9%	191/399	6257	4.7%	111/180	0.019
	Multivitamins	8018	23,099	25.7%	2236/3707	10,668	19.4%	905/1168	<0.001
	Fish oil	3139	10,317	21.3%	894/1306	4581	20.5%	454/485	0.263
	Calcium	3978	16,983	16.4%	413/2367	7345	16.3%	232/966	0.924
	Omega 3	797	13,468	4.2%	193/373	6129	3.8%	99/132	0.166
	Saw palmetto	110	12,835	0.5%	-	6043	0.7%	-	0.135
	Sildenafil	218	21,168	0.7%	-	9221	0.9%	-	0.068
	Zinc	394	14,823	1.9%	-	6208	1.8%	-	0.518
	Tadalafil	108	21,571	0.3%	-	9461	0.4%	-	0.454
	Flax seed	409	9780	2.9%	-	4389	2.8%	-	0.573
	Folic acid	975	9887	6.5%	-	4430	7.6%	-	0.019
	Garlic pills	229	13,082	1.2%	--	6284	1.1%	-	0.798
	Ginger pills	45	5957	0.4%	-	6268	0.3%	-	0.286
	Ginko	167	13,066	0.9%	-	6279	0.9%	-	1.000
	Ginseng	62	13,038	0.3%	--	6269	0.4%	-	0.145
	Glucosamine	1262	13,337	6.4%	-	6417	6.4%	-	0.928
	Iron	1513	13,386	8.3%	-	6402	6.4%	-	<0.001
	Magnesium	1456	22,088	4.7%	-	10,260	4.2%	-	0.055

* Chi-square test of independence.

**Table 2 medicina-59-00919-t002:** Supplement consumption prevalence in periodontal health versus periodontitis and results of Chi-Square test of independence.

	Conditions	Total (*n*)	Periodontal Health/Gingivitis (*n*)	Percent (+) for Supplement Consumption with Periodontal Health/Gingivitis	Periodontitis (*n*)	Percent (+) for Supplement Consumption with Periodontitis	*p*-Value *
Periodontal status only			85,809	-	32,715	-	-
Dietary Supplements	Vitamin B	2961	16,796	12.3%	7027	12.8%	0.320
	Vitamin C	2138	15,217	10.1%	6341	9.5%	0.243
	Vitamin D	6751	16,056	29.4%	6682	30.3%	0.196
	Vitamin E	881	14,991	3.9%	6257	4.7%	0.019
	Multivitamins	8018	23,099	25.7%	10,668	19.4%	<0.001
	Fish oil	3139	10,317	21.3%	4581	20.5%	0.263
	Calcium	3978	16,983	16.4%	7345	16.3%	0.924
	Omega 3	797	13,468	4.2%	6129	3.8%	0.166
	Saw palmetto	110	12,835	0.5%	6043	0.7%	0.135
	Sildenafil	218	21,168	0.7%	9221	0.9%	0.068
	Zinc	394	14,823	1.9%	6208	1.8%	0.518
	Tadalafil	108	21,571	0.3%	9461	0.4%	0.454
	Flax seed	409	9780	2.9%	4389	2.8%	0.573
	Folic acid	975	9887	6.5%	4430	7.6%	0.019
	Garlic pills	229	13,082	1.2%	6284	1.1%	0.798
	Ginger pills	45	5957	0.4%	6268	0.3%	0.286
	Ginko	167	13,066	0.9%	6279	0.9%	1.000
	Ginseng	62	13,038	0.3%	6269	0.4%	0.145
	Glucosamine	1262	13,337	6.4%	6417	6.4%	0.928
	Iron	1513	13,386	8.3%	6402	6.4%	<0.001
	Magnesium	1456	22,088	4.7%	10,260	4.2%	0.055

* Chi-square test of independence.

## Data Availability

Dara is available upon request.

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
