# Peer review of "Supplement Consumption and Periodontal Health: An Exploratory Survey Using the BigMouth Repository"

_medicina, 2023, doi:10.3390/medicina59050919_

Round 1

Reviewer 1 Report

Dear Authors,

I have completed my evaluation of the manuscript.

These are my comments:

1) The manuscript is well written.

2) Regarding periodontal health and periodontitis, the study provided satisfactory results.

3) As a result, 1) multivitamins and iron, and 2) folic acid and vitamin E are associated with periodontal health and periodontitis, respectively.

4) Overall, the present study reported a minimal association between periodontal health and the consumption of dietary supplements.

5) Thus, the manuscript can be accepted for publication.

Author Response

We wish to thank the author for his/her review and kind words. 

Reviewer 2 Report

First of all, I would like to thank the Editor-in-Chief and Associate Editors from Medicine journal for giving me the opportunity to have reviewed the Manuscript ID: Medicina-2252213 titled “Supplement Consumption and Periodontal Health: Na Exploratory Survey using the BigMouth repository”. The main objective of the present study is to examine the correlation between populations who report taking different dietary supplements and their relative periodontal health.

The idea of the study is interesting. although it points out many limitations of writing and the methodology is not clear

Below, I have some questions for the authors

In the abstract, in the methods. the authors indicate evaluating the prevalence of periodontitis in comparison with health periodontal disease in all patients with systemic diseases. What systemic diseases were included or evaluated?

Still, in the abstract results, the authors do not indicate what was the prevalence of periodontitis compared to health periodontal disease in patients with systemic diseases.

In conclusion, in the abstract, the authors indicate that they found a minimal association between periodontal health and the consumption of dietary supplements, but the results do not specify which supplements had this association. Specify which supplements had a limited membership.

Please see my specific comments:

P1 paragraph 1. Make this first paragraph more succinct.

Talk more about supplements in the introduction, such as talking about the impact of omegas 3 and 6 on periodontics, calcium, or the effect of adjunctive periodontal therapy with food supplementation

.

P2 paragraph 2. “Many authors have hypothesized... but do you not cite the references of the studies?

P2. “Supplementation studies, provide an important assessment of the systemic landscape that influences….” Indicate the references of the studies, and include a paragraph about what they studied. Also, how can this study help?

The authors only mention 3 references for the entire introduction. The introduction of the article is very limited.

Methods

In methods, 2.1/ in predetermined definitions: According to the codes, specify the definition of periodontal health, gingivitis, and periodontitis.

So also indicate who carried out the periodontal diagnosis of the patients, was it a general practitioner or a specialist in the area?

Dental terminology codes were used... CDT codes are used to code dental procedures. Please specify which periodontal procedures were performed on the patients included in the sample.

2.2 Sample selection

Indicate whether there was an age range within the general eligibility criteria

The general eligibility criteria for the sample were adult dentate patients receiving preventive or periodontal treatment.

did these patients undergo periodontal treatment adjuvant to supplements?

In the summary it indicates that they evaluated the prevalence of periodontitis in comparison with periodontal health, in all patients with systemic diseases, indicating within the eligibility criteria of the patients included in this study, what type of systemic diseases did they have. Please specify.

Also, indicate whether smokers and/or pregnant patients were included.

2.4. In the statistical analysis, explain which dependent and independent variables the authors used. to explain better

In the statistical analysis, explain which dependent and independent variables the authors used. to explain better. And what software and statistics program was used?

Results

In the results, in sample characteristics, what was the average age of these included patients?

The authors indicate in the abstract that the prevalence of periodontitis was evaluated in comparison with periodontal health in all patients with systemic diseases. Do the results still not indicate which systemic diseases are evaluated and the environmental factors?

These data are important because depending on the type of systemic disease, systemic disease can influence periodontal health. Even so, some risk factors such as smoking.

In conclusion

Specify which supplements had a limited membership.

Reviewer 3 Report

General Comments

The authors examined the relationships between supplement intakes and periodontal health/diseases using their big data, the University of Michigan database via the BigMouth repository. They found interesting correlations, especially multivitamins, iron, vitamin D, and folic acid towards periodontal health/disease, while they understand well regarding the limitations of this study. Although this is an intriguing finding, the analysis and the results should be evaluated more appropriately, and the manuscript needs more detailed descriptions to support their conclusion. The following improvements should be considered.

Major Comments

I. The number of subjects (118,524) does not match the total number (119,426) of male (55,459) and female subjects (62,967). In addition, each parameter in Tables and the formula of percent (+) for Supplement Consumption with Periodontal Health/Gingivitis or Periodontitis were not clear in this manuscript. Please add more information for each parameter in Tables and address how to calculate them in the Material and Methods section.

II. Tables 1 and 2 look almost the same except for gender predilection, while there was no detailed description regarding gender in the Results section. The authors need to describe gender differences in supplement intake and their effect on the current results in the Result section.

III. Since it seems that quite a few subjects are taking more than one dietary supplement, it is important to consider multiple regression analysis rather than the Chi-square test of independence.

Minor Comments

i.  It may be a typo but please double-check with “--” in Table 1 if they were correct.

ii. The sentence “Summarizing that nutrition factors in terms of nutrient intake ~” in the first paragraph of the Introduction seems to come out of nowhere while there is no mention of nutrients before or after in this paragraph. Please review if this is the correct story or maybe refer to more appropriate references for the biology of Periodontitis and its immune response in the first paragraph.

Round 2

Reviewer 2 Report

En primer lugar, me gustaría dar las gracias al Editor en Jefe y Editores Asociados de Medicina por darme la oportunidad de tener revisó la identificación manuscrita: medicina- 2252213 titulada "Suplemento de consumo y periodontal" Salud: Una encuesta exploratoria utilizando el repositorio de boca grande".

El documento ha mejorado desde la última revisión, pero usted todavía puede mejorar la introducción del manuscrito.

Introducción

Primer párrafo de la introducción

Si bien hay un creciente cuerpo de investigación que explora El impacto de estos suplementos y superalimentos en la salud humana y su posibles aplicaciones médicas. Añadir más referencias

Párrafo 2: añadir más referencias

Párrafo 4

Los estudios de suplementación proporcionan un importante evaluación del panorama sistémico que influye en las enfermedades inflamatorias, incluyendo periodontitis. Agregar referencia

Referencias

Coloque las referencias de acuerdo con el alcance de la diario.

First of all, I would like to thank the Editor-in-Chief and Associate Editors of Medicine for giving me the opportunity to have reviewed the handwritten identification: medicine- 2252213 titled "Consumable and Periodontal Supplement" Health: An exploratory survey using the mouth repository big".

The document has improved since the last revision, but you can still improve the manuscript introduction.

Introduction

First paragraph of the introduction

While there is a growing body of research exploring the impact of these supplements and superfoods on human health and their potential medical applications. Add more references

Paragraph 2: add more references

Paragraph 4

Supplementation studies provide an important assessment of the systemic landscape influencing inflammatory diseases, including periodontitis. add reference

References

Place the references according to the scope of the journal.
